

# Stigmatization towards healthcare personnel during the first COVID-19 wave in Central and Northern Mexico

Christian Enrique Cruz-Torres[1] and Jaime Martín del Campo-Ríos[2]

[1] Departamento de Psicología, Universidad de Guanajuato, León, Guanajuato, Mexico
[2] Instituto de Ciencias Sociales y Administración, Universidad Autónoma de Ciudad Juárez, Ciudad Juárez, Chihuahua, México

## ABSTRACT

The evidence all over the world shows an alarming increase in the stigmatization of health personnel during the COVID-19 pandemic. We sought to explore possible psychological factors that help explain the disposition to stigmatize health personnel in the central and northern regions of Mexico. Two studies explore possible psychological factors to explain the disposition to stigmatize healthcare personnel (HP) in Mexico during the COVID-19 pandemic. In study one, 520 participants responded to three instruments that measure the disposition to stigmatize, the perceived contagion risk, and the positive beliefs towards HP. Results showed a generalized low disposition to stigmatization, where only a small percentage obtained high scores. A regression analysis identified that stigmatization towards HP can derive mainly from the perception of risk of contagion, although positive beliefs of HP decrease this disposition. The second study extends this finding by analyzing responses of 286 participants to seven instruments measuring factors hypothesized as predictors towards stigmatization: uncertainty generated by the pandemic, selfish strategies to face off the pandemic, social capital, trust in institutions, perceived vulnerability of contagion, perceived risk of contagion, and positive beliefs towards HP. A path analysis reveals that the main predictor of stigmatization is the perceived risk of contagion, increased by the strategy of selfishness, and the uncertainty generated by the pandemic. These results are discussed emphasizing the importance of cooperation and community ties to prevent the stigmatization of HP in the context of sanitary emergencies generated by contagious diseases.

# INTRODUCTION

Healthcare personnel (HP) from all around the world located in the first lines of defense during the COVID-19 health contingency, were one of the most vulnerable sectors to marginalization (*Bhanot et al., 2021*). In Mexico, the mistreatment of medical personnel escalated to the point of being threatened, being attacked with hot coffee, eggs, and other verbal and physical attacks (*Semple, 2020*). In April 2020, less than a month after the World Health Organization (WHO) declared COVID-19 a pandemic, at least twenty-one complaints from health workers and close to one hundred and forty calls related to

Corresponding author
Jaime Martín del Campo-Ríos,
jaime.martin@uacj.mx

acts of discrimination taken for one hour were registered with the National Council to Prevent Discrimination in Mexico. This was equivalent to what they typically received in a week (*González Díaz, 2020*). These attacks on health personnel have occurred in different countries during other epidemics and also now in the COVID-19 pandemic (*Yuan et al., 2021*), and can find their explanation in the fear of being infected, but they require a deeper analysis since they violate human rights and obviously, harming those who care for our health is extremely detrimental to common well-being, especially when we face a health emergency such as that caused by COVID-19. The concepts of prejudice and stigmatization are analyzed in the present project given that we aim to provide evidence of both as fundamental in explaining aggressions toward HP.

## Prejudices, stigma, and fear of contagion

Prejudices have been traditionally understood as negative emotional responses oriented towards members of stable and well-delimited social groups in their relation to other social groups. For instance, *Dovidio et al. (2010)* defined prejudice as an individual attitude that may have a subjective, positive, or negative tone toward a group and its members that creates or maintains unequal hierarchical relations between their members. Meanwhile, *Stangor (2016)* defines prejudices simply as a negative attitude towards a group and its members. Although prejudices have been studied widely to understand discriminatory related phenomena such as racism, *Schaller & Neuberg (2012)* recur to the evolutionary roots of prejudice highlighting its function of evading two potential common threats to any human group in its evolutionary story: violence from other human groups, and the risk of contracting infectious diseases. From this latter perspective, racism and in-group bias would represent the defensive responses to the potential threats of violence from other groups. From the former perspective, *Schaller (2015)* proposes the existence of an immune behavioral system that will aid in identifying current traits in our social environment that will suggest the risk of contracting infectious diseases to avoid them in a timely manner. For instance, *Makhanova and Sheperd*'s *(2020)* results showed that a major perceived vulnerability toward diseases was associated with major social distancing actions during the COVID-19 pandemic.

The behavioral immune system can increase the salience of prejudices, for example, the results of *O'Shea et al. (2020)* show that racial prejudices, both implicit and explicit, are higher in communities that had a major prevalence of infectious diseases, supporting the hypothesis of prejudices as a cognitive bias that evolved as a mechanism to maintain the individuals away of potential violence threats and diseases. In a similar way, the results of *Lu et al. (2021)* showed that prime COVID-19 salience increases prejudices and the intention to discriminate against individuals of Asian and Hispanic ethnicity.

The rejection of those who potentially carry disease and imply a contagion risk has been also studied under the concept of stigma, understood as the devaluation and exclusion of some individuals in society based on visible characteristics associated with the risk of contagious diseases (*Bhanot et al., 2021*). *Phelan, Link & Dovidio (2008)* analyzed the concepts of prejudice and stigma and found that both concepts are highly similar in their definitions, although they have been applied in the explanation of different

phenomena. Meanwhile, the concept of prejudice has been mainly applied to the analysis of discrimination by ethnical and racial motives, stigma has been focused on the study of *deviations*, like identities and behaviors that transgress traditional social norms, and to the study of discrimination by disabilities and diseases. In this sense, the exclusion and violence that some HP elements have experienced are closer to the concept of stigma, by being motivated by marginalizing them to avoid a contagion risk.

From this approach, the revision of *Baldassarre et al. (2020)* shows that rejection of potentially sick persons has been witnessed in epidemics of diseases such as HIV, tuberculosis, and Zika. In the same vein, *Bhanot et al. (2021)* show that the stigma of COVID-19 was combined in India with the already existent prejudices against some of the groups discriminated by their ethnic condition, religion, or migratory status, exacerbating their previous problems of discrimination. Their results also identify HP as a discriminated sector of society for an alleged higher risk of contagion compared to the rest of the population.

Based on these antecedents, two studies were carried out. The first study sought to quantify the disposition of the population in Mexico to marginalize HP and to identify if this disposition was associated with the perception of HP as a possible risk of contagion. The second study extends these results by analyzing other explanatory factors of marginalization towards HP in a second Mexican sample.

## Study 1. Descriptive and Sociodemographic Components of stigmatization

Cases of violence and rejection towards HP have been reported in Mexico under the argument of implying a risk of contagion (*Semple, 2020*; *González Díaz, 2020*), but there are no studies that analyze the perception of the general population towards HP. This first study explores the perception of a sample of Mexican inhabitants towards HP, in terms of being positive, being perceived as a risk of contagion, and the disposition to marginalize them socially. Considering the isolated reports of violence, and assuming a widespread fear in the population of a disease that has cost the lives of millions, it can be proposed that HP, who are exposed daily to the virus more than others, are possibly perceived as a threat to society, due to an assumed higher capacity to spread the virus. At the same time, the important work of HP caring for community people against COVID-19 can generate a positive perception in the population, which would protect them from being marginalized. To test this hypothesis, a quantitative, cross-sectional, correlational study, with an explanatory scope, was carried out.

## MATERIALS & METHODS

### Participants

Participants were 193 (34.2%) men, 333 (58.9%) women, and 39 (6.9%) that not answer that question, aged between 17 and 68 years ($M = 24.08$, SD $= 7.62$), residents of northern (76%) and central-southern states (24%) from Mexico. 42.2% declared having unfinished careers, 34.2% upper secondary studies, and 19% completed undergraduate studies. 2.9% reported working in a hospital and 29.2% declared having relatives who worked in a

hospital. 74.5% stated that they did not have children. No one reported having been diagnosed with COVID-19 up to the time of the survey and 92.9% confirmed that they had not had related symptoms. Only 2.8% stated that one of their family members had been diagnosed with COVID-19 and 84.4% stated that no one in their family had experienced related symptoms.

## Instruments

*Marginalization towards healthcare personnel.* It is made up of six items: (1) If I had a neighbor who works in a hospital, I would prefer not to find him on the street in order to not get infected; (2) Even if I could help a doctor or a nurse, I would prefer not to do it so as not to risk getting infected; (3) The children of nurses and doctors should not be admitted to nurseries because they can infect other children; (4) Staff working in hospitals should be prevented from using public transport to avoid infecting others; (5) If a person working in a hospital asked me for help I would prefer not to do so in order to avoid being infected; (6) It would be best if the doctors and nurses moved near the hospitals in order to avoid infecting others. The exploratory factor analysis identified a single factor that groups the six items and explains 52% of the variance with Cronbach's alpha index = .85.

*Perceived contagion risk towards healthcare personnel.* It is made up of three items: (1) If I am buying something and a doctor or a nurse arrives at the same place, I would worry that they could infect me; (2) If a doctor or a nurse is on public transport as me, I would be afraid of being infected by them; (3) Being close to a doctor or a nurse implies a higher risk of contagion than people who do not work in the medical industry. The exploratory factor analysis identified a single factor that groups the three items, explaining 62% of the variance with Cronbach's alpha = .80.

*Positive beliefs towards healthcare personnel.* It is made up of six items: (1) Faced with this contingency, people who work in hospitals are risking their lives for the good of everyone; (2) Nurses and doctors are the ones who most deserve our support in this contingency; (3) Doctors and nurses are acting with great courage at work since they are most at risk of infection; (4) If I could support the doctors and nurses in this contingency, I would gladly do so; (5) At the end of this contingency, we will all be in debt to the country's doctors and nurses; (6) While we stay at home, doctors and nurses risk their lives to help others. The exploratory factor analysis identified a single factor that groups the six items, explaining 39.8% of the variance with Cronbach's alpha index = .77.

Responses to these instruments were rated on a Likert-type scale ranging from 1 (Strongly disagree) to 4 (Strongly agree). In addition, it required sociodemographic data such as age, sex, educational level, whether they or a relative worked in a health care center, whether they had children and whether they or their relatives had received a positive diagnosis for COVID-19, and the state of residence.

## Procedure

The Autonomous University of Juárez City granted full ethical approval to conduct the study (Ethical Permission Reference: CEI-2020-2-43). Participants were invited to participate in the study via email containing a link to the study website. Measures were

administered through the SurveyMonkey online tool (SurveyMonkey, San Mateo, CA, USA; http://www.surveymonkey.com).

The survey was conducted from the second to the fourth week of April 2020, one month after the WHO declared COVID-19 a pandemic on March 11, 2020 ("Coronavirus confirmed as a pandemic", 2020), three weeks after essential face-to-face activities were partially or totally abolished in Mexico on March 26, 2020 (*Palma, Rubio Barnetche & Lecona, 2020*), and one week after a national health emergency was declared in Mexico on March 31, 2020 (*Borunda, 2020*). The support of students and acquaintances was requested to invite possible full-time workers as participants. If they agreed to participate, the details of the informed consent and the procedures for completing the measures were explained to them.

In order not to expose the health of the participants during the quarantine period, they were reminded that these invitations should be made electronically, without leaving their homes. With these characteristics, the sampling used in this study is considered non-probabilistic. Consent was obtained by digital means from all participants. They were informed that their answers would be confidential, their information would be protected by the research team and their participation would be voluntary.

## Data analysis

The construct validity of the instruments was verified by exploratory factor analysis with the maximum likelihood extraction method, with an eigenvalue greater than 1 as an extraction criterion. The internal consistency of each factor was calculated using Cronbach's alpha formula. Once the structure and internal consistency were verified, new indicators were formed for each instrument by averaging their items. Mean comparisons were performed using t-tests and one-way analysis of variance using the software Jamovi (*The jamovi project, 2021*). To verify the hypotheses of predictive effects on marginalization, multiple linear regressions were performed using the stepwise method in the SPSS 22 program (*IBM Corp, 2013*).

## RESULTS

As seen in Fig. 1, the averages of marginalization and perceived risk are generally low, nearby to the response options "Totally disagree" and "Disagree", while the average of positive perceptions is located closer to the "Totally agree" option. These would be the general trends, but it is identified that 5% report average scores of marginalization between 2.5 and 4, that 10% report average scores between 3 and 4 of the perceived risk of contagion, and that 5% report scores of 3 and lower of positive beliefs towards HP.

Table 1 shows the comparison of marginalization averages through the different socio-demographic indicators. Statistically significant differences are observed between those who have or do not have family members who work in a health care center, with slightly higher scores on marginalization in those who do not have family members working in these centers. Those who reside in the north of the country also report slightly higher scores than the central-southern states. In both cases the scores do not reach the value 2,

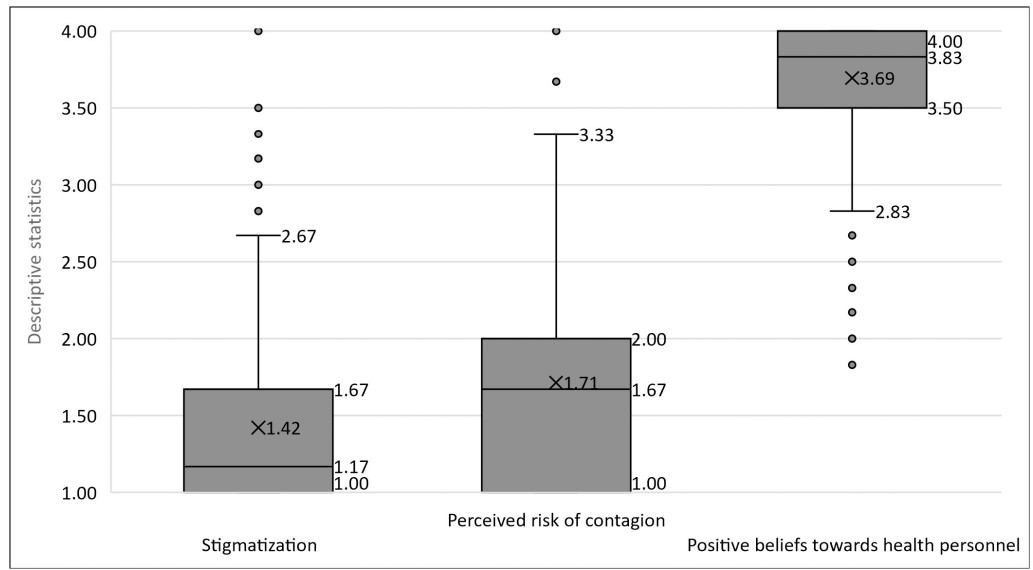

**Figure 1** **Descriptive statistics of stigmatization, perceived risk of contagion, and positive perceptions towards health personnel.** Low scores are observed for stigmatization and perceived risk of contagion, and high scores for positive perceptions towards health personnel.

**Table 1 Comparison of the averages of stigmatization towards health personnel by different sociodemographic indicators.**

| Variable | Statistical result | Group | Mean |
|---|---|---|---|
| Some relative Works at a clinical or hospital | $t_{332.69} = -2.12, p = .03, d = -.19$ | Yes | 1.35 |
| | | No | 1.45 |
| Country zone | $t_{224.90} = 2.23, p = .02, d = .25$ | North | 1.43 |
| | | Center-South | 1.32 |
| Works at a clinical or hospital | $t_{522} = -1.71, p = .08, d = -.48$ | Yes | 1.20 |
| | | No | 1.43 |
| Sex | $t_{524} = .56, p = .57, d = .05$ | Men | 1.44 |
| | | Women | 1.41 |
| Having children | $t_{593} = -.197, p = .84, d = -.02$ | Yes | 1.41 |
| | | No | 1.43 |
| Level of schooling | $F_{4,521} = 2.30, p = .05$ | Primary | 1.46 |
| | | High school | 1.42 |
| | | Bachelor uncomplete | 1.48 |
| | | Bachelor degree | 1.32 |
| | | Postgraduate | 1.27 |

**Notes.**
Source: Own elaboration.

indicating an opinion against marginalization. Cohen's d with values close to .2 indicates a small effect size for both differences.

The regression analysis showed positive effects of the perceived risk of HP (B = .44, $\beta$ = .61, t = 18.95, p < .001) and negative effects of positive beliefs towards HP (B

$= -.15$, $\beta = -.11$, $t = -3.57$, $p < .001$), which together explain 40% of the variance of marginalization towards HP ($R^2 = .40$, $F_{2,562} = 189.03$, $p < .001$). With a tolerance level $= .99$, collinearity problems between the independent variables are discarded.

## Conclusions of study 1

The social perception of HP can be considered positive, with low scores of marginalization and perceived risk of contagion and high scores of positive beliefs. Slightly higher scores of marginalization are identified in those who do not have relatives working in healthcare centers and inhabiting the northern region of the country. Although these scores are low, indicating a rejection of beliefs of marginalization towards HP. However, it should be noted that a low percentage reported high scores for disposition to marginalization and perceived risk of contagion toward HP. The regression analysis identifies that marginalization towards HP can derive mainly from the perception of risk of contagion, although the beliefs of HP as heroes who risk their lives for the good of society decrease the disposition to marginalization derived from the perceived risk of contagion.

## Study 2: psychosocial predictors of stigmatization towards HP

Study 1 showed a generalized low disposition to marginalization in most of the population, although a small percentage did report this disposition in high scores. It was also identified that the perceived risk of contagion is an important predictor of marginalization, while positive beliefs towards HP help to diminish this effect. Given these results, it is necessary to identify some factors associated with a greater disposition to marginalization to understand this phenomenon.

Based on the previous findings, this second study proposes the exploration of the following as explanatory factors of marginalization toward HP.

## Cooperation

Cooperation is understood as a practice where an individual or group invests part of their resources (*e.g.*, time, money, work) in a joint task with another individual or group to obtain a common benefit (*Bowles & Gintis, 2011*). This investment always involves some risk that the other investors betray our trust, for example, not contributing their resources hoping that the investments of others were sufficient, or appropriating the obtained benefits and not sharing them.

Attacks on HP or ethnic groups under the argument that they imply a risk of contagion may be indicating a tendency to reserve cooperation only for the closest members of our group. For example, *Strachman & Schimel (2006)* argued that thinking about the possibility of dying motivates the need to defend a general vision of how the world works according to our own beliefs, showing evidence that generating thoughts about one's own mortality leads to a lower commitment to the romantic partner, but only when both individuals endorse very different beliefs. Using a similar methodology, *Renkema et al. (2008)* showed that people induced to think about their own death were more likely to change their own ideas and adhere to ideas common in their own group but rejected ideas coming from different groups. In addition, they tended to perceive people from other groups based on stereotypes, without dwelling on their differences, which can lead to a greater perception of threat from

the group and its members (*Haner et al., 2020*). This behavior would be explained as a psychological strategy that would favor stronger alliances by motivating the formation of more heterogeneous groups that would allow them to confront a possible death threat. These individual cognitive processes can lead to the decomposition of the broader social fabric, affecting bonds of trust and reciprocity fundamental to the well-being of more heterogeneous communities, motivating individuals and communities to lock themselves in their closest social nuclei, deny wider cooperation, and escalate the level of hostility towards others, in this case towards health care personnel.

## Uncertainty

Another factor that can exacerbate violence against others is the uncertainty generated by the pandemic. *Brizi, Mannetti & Kruglanski (2016)* found that people with a dispositional need to find a quick response to situations of uncertainty, known as a need for closure, tended to discriminate more frequently against people from other groups. However, this tendency for discrimination was equally increased when uncertainty was intensified through experimental manipulation, even in individuals with lower levels of need for closure. That is, uncertainty, whether due to a personality disposition or generated by external conditions (*e.g.*, a pandemic), increases the tendency to discriminate against those who are perceived as different. *Cruz-Torres & Martín del Campo-Ríos (2022)* identified that the uncertainty generated by the pandemic increases the disposition to selfishness (*e.g.*, believing that during the contingency seeing for others is a mistake) and the perceived selfishness in others (*e.g.*, considering that with contingency people try to get what they want, even going over others).

## Social capital

These effects of uncertainty on cooperation may be less important in communities that have stronger bonds of reciprocity and trust. In this sense, *Nanetti, Leonardi & Putnam (1994)* propose that communities vary in their levels of social capital, which is defined as the concordance between social trust, norms of reciprocity, and networks of civic commitment in an association of people to coordinate collective actions. Thus, communities that maintain their networks after successfully becoming organized to solve common problems, trust each other and keep their bonds active through reciprocal exchanges, are said to have high social capital.

These resources of the community have been related to a higher perception of safety, for example, in the face of criminal violence (*Hansen-Nord et al., 2014*; *Dinesen et al., 2013*). In the case of the sanitary crisis, the results of *Gonzalez-Medina & Le (2011)* show that a higher prevalence of infectious diseases is associated with lower levels of interpersonal trust, which can lead to a deterioration of the social fabric. In the same sense, *Baldassarre et al. (2020)* show that the stigmatization of potentially sick persons during epidemics has also implied the social fabric diminishment of the communities. After considering this capacity, higher levels of social capital can be expected to be associated with a lower disposition to non-cooperation and the marginalization of HP.

### Perceived vulnerability to contagion

Given that the root of uncertainty, no cooperation, and margination is the fear of contagion, it is likely that people who perceive themselves to be especially susceptible to contagion tend to present greater fear and uncertainty, and with it, more intense selfishness and disposition to marginalize others. In this regard, *Duncan, Schaller & Park (2009)* have shown that the perceived vulnerability to contagion can be considered an individual difference and that people have higher or lower levels that can be quantified psychometrically. *Mallett et al. (2021)* showed that perceived vulnerability to contagion and intolerance of uncertainty are associated with greater anxiety during the pandemic. In the same sense, *Padmanabhanunni et al. (2022)* demonstrated that those who report high levels of perceived vulnerability to contagion have suffered more anxiety, depression, and hopelessness during the pandemic. These antecedents motivate further exploration of the hypothesis that higher levels of perceived vulnerability to contagion are associated with a greater perception of the risk of contagion of HP and a greater willingness to marginalize them.

In summary, the study conducted by *Cruz-Torres & Martín del Campo-Ríos (2022)* proved that the uncertainty generated by the pandemic increased strategies of selfishness in the community, an effect that was diminished in those who perceived that their community had bonds of reciprocity, interpersonal trust, and civic engagement, which are all components of social capital. In turn, the measurement of *Duncan, Schaller & Park (2009)* makes it possible to identify variations in the perceived vulnerability to contagion, a factor that could increase the effects of uncertainty and the perceived risk of contagion on marginalization towards the HP. Finally, a factor that cannot be ignored is the trust in government and health institutions, which are elements that can help prevent violence against HP.

Considering this background, this second study aims to explore the effects of the uncertainty generated by the pandemic, selfish strategies, social capital, trust in institutions, perceived risk of contagion, positive beliefs towards HP, and the perceived vulnerability of contagion on the willingness to marginalize HP in a sample of Mexican inhabitants. It is proposed as a hypothesis that the uncertainty generated by the pandemic, the perceived risk of contagion, the perceived vulnerability of contagion, and selfishness will increase the willingness to marginalize HP, while positive ideas towards HP, the components of the social capital and trust in health and state authorities will help reduce this disposition.

## MATERIALS & METHODS

### Participants

Participants consisted of 110 men (38.5%), 176 women (61.5%), and two people that do not answer that question, aged between 18 and 63 years ($M = 23.98$, SD $= 7.57$), residents of northern (79%) and central-southern (21%) states of Mexico. Regarding the educational level, 45.8% had unfinished undergraduate studies, 17.9% had intermediate-level studies and 26.9% had completed undergraduate studies. A total of 1.8% reported working in a hospital and 25.7% declared having relatives who worked in a hospital; 82.6% stated that they did not have children. No one reported having been diagnosed with COVID-19 and

95.1% stated they had not had related symptoms. A total of 7.7% stated that one of their relatives had been diagnosed with COVID-19 and 78.2% stated that no one in their family had experienced symptoms.

## Instruments

The same instruments used and described in study 1 were used for this study, besides the following measurements.

*Community Assessment of Social Capital* (*Cruz & Contreras-Ibáñez, 2015*). Responses are measured in 10 items on a 4-point Likert-type scale, from 1 (Strongly disagree) to 4 (Strongly agree). The reciprocity factor refers to the willingness to support and the expectation of being reciprocated (*e.g.*, If a neighbor asks me for a favor, I know that I will have their support when I need it). The second factor is civic engagement networks, which refers to the ability and willingness of neighbors to organize and solve community problems (*e.g.*, if a problem arose on our streets, the neighbors would organize quickly). Finally, the distrust factor refers to these negative beliefs toward neighbors (*e.g.*, If I am careless, my neighbors would take the opportunity to do something bad to me). The scores of these elements were recorded inversely, so the factor was named confidence. Cronbach alpha values were above >.80 for each factor (Cruz & Contreras, 2015).

*Strategies of selfishness during the pandemic.* With three items, its factor selfishness measures the concentration of cooperation during the pandemic in the closest social circles, (*e.g.*, In these moments of contingency it is best to see for your family, not for others). The second factor, perceived selfishness, measures the perception that others are not willing to cooperate either with three items (*e.g.*, During a health contingency people try to see only for their own benefit). Responses are rated on a Likert-type scale from 1 (Strongly disagree) to 4 (Strongly agree). Confirmatory factor analysis showed adequate goodness-of-fit, and Cronbach alpha values above .7 for each factor (*Cruz-Torres & Martín del Campo-Ríos, 2022*).

*Measurement of the uncertainty resulting from the coronavirus contingency.* Adapted from *Lambert et al. (2014)*, this instrument measures the perception of uncertainty in the face of changes derived from the health contingency (*e.g.*, At this time I am not sure of my ability to successfully face this contingency) using a Likert scale from 1 (Strongly Disagree) to 4 (Strongly Agree). Confirmatory factor analysis showed a single factor grouping its five items with adequate goodness-of-fit indices and Cronbach alpha values above .7 for each factor (*Cruz-Torres & Martín del Campo-Ríos, 2022*).

*Perceived vulnerability to disease* (*Duncan, Schaller & Park, 2009*). The 7-item Perceived Infectivity subscale examines individuals' beliefs about their susceptibility to infectious diseases (*e.g.*, In general, I am very susceptible to colds, the flu, and other infectious diseases). The germ aversion subscale (eight items) measures people's discomfort in situations that connote a higher probability of transmission of pathogens (*e.g.*, I prefer to wash my hands soon after shaking someone's hand). Both subscales were answered in a Likert scale from 1 (Strongly Disagree) to 4 (Strongly Agree). Confirmatory factor analysis showed adequate goodness-of-fit indices with Cronbach alpha values above .7 for each factor (*Cruz-Torres & Martín del Campo-Ríos, 2022*).

*Trust in institutions.* Trust towards two institutions was evaluated through two independent items: (1) "How much do you trust the health authorities of your state?" and (2) "How much do you trust the governor of your state?", both presented in a Likert-type format with response options ranging from 1 (not at all) to 5 (a lot).

### Procedure

The same procedure and ethical care described in study one was followed. The survey was carried out from the last week of May and the first week of June 2020.

### Data analysis

To verify the hypotheses of predictive effects on marginalization, multiple linear regressions were used using the stepwise method in the SPSS 23 program. To integrate the effects of the independent on the dependent variables in a single model, a trajectory analysis was carried out with the AMOS 22 program (*Arbuckle, 2013*).

## RESULTS

As in study 1, the averages of marginalization (1.4) and perceived risk of contagion (1.69) were low and positive beliefs towards HP were high (3.17).

The hypotheses of the effects of the independent on the dependent variables were verified by means of linear regressions before proceeding to the trajectory analysis. The regression on marginalization towards HP confirms the effects found in study 1 of the risk of contagion and positive ideas towards HP, adding the effects of selfishness. The model explains 52% of the variance of marginalization ($R^2 = .52$, $F_{3,284} = 106.18$, $p < .001$) derived from positive effects of the risk of infection of HP ($B = .39$, $\beta = .58$, $t = 13.98$, $p < .001$), selfishness ($B = .18$, $\beta = .27$, $t = 6.52$, $p < .001$) and negative effects of positive ideas towards HP ($B = -.16$, $\beta = -.10$, $t = -2.57$, $p = .01$). The tolerance levels obtained higher than .93 rule out problems of collinearity between the independent variables.

Subsequently, the effects of regression towards the risk of contagion perceived by HP were explored, having as independent variables the factors of social capital (reciprocity, civic engagement networks, and trust), trust towards state health authorities, trust towards the governor of the state, the uncertainty in the face of COVID-19, their selfishness and the selfishness perceived in others. The model explains 9% of the variance of the risk of contagion perceived by HP ($R^2 = .09$, $F_{3,281} = 9.25$, $p < .001$) derived from the positive effects of selfishness ($B = .21$, $\beta = .21$, $t = 3.63$, $p = .01$), the uncertainty generated by the COVID-19 pandemic ($B = .13$, $\beta = .13$, $t = 2.39$, $p = .01$) and negative effects of confidence in the state health authorities ($B = -.12$, $\beta = -.13$, $t = -2.39$, $p = .01$). Tolerance showed scores higher than .96, discarding collinearity problems. The factors of social capital, trust in the governor, perceived selfishness in others, aversion to germs, and contagion vulnerability did not show statistically significant regression coefficients and were excluded from the model.

The same variables, plus the perceived risk of contagion from HP, were used as independent variables to predict the positive beliefs about HP. The resulting model explains 7% of the variance ($R^2 = .078$, $F_{3,280} = 7.73$, $p < .001$) derived from the positive

effects of trust in health authorities (B = .10, $\beta$ = .21, $t$ = 3.63, p<.001) and selfishness perceived in others (B = .10, $\beta$ = .16, $t$ = 2.78, $p$ = .006) and negative effects of the selfishness factor (B = −.10, $\beta$ = −.16, $t$ = −.28, $p$ = .005). The tolerance was greater than .90, discarding collinearity problems in the model. The factors of social capital, trust in the governor, germ aversion, contagion vulnerability, and uncertainty did not show statistically significant regression coefficients and were excluded from the model.

The analysis was also replicated to predict selfishness, finding positive effects of repeated perceived selfishness in others (B=.28, $\beta$ = .29, $t$ = 5.32, $p$ < .001), the perceived risk of HP (B = .18, $\beta$ = .19, $t$ = 3.45, $p$ = .001), trust in state authorities (B = .10, $\beta$ = .14, $t$ = 2.50, $p$ = .01), and negative effects of positive beliefs towards HP (B = −.21, $\beta$ = −.13, $t$ = −2.38, $p$ = .01). Together, these variables explain 15% of the variance of selfishness ($R^2$ = .15, $F_{4,279}$ = 12.86, $p$ < .001), discarding collinearity problems with tolerance values greater than .93. The factors of social capital, trust in the governor, germ aversion, contagion vulnerability, and uncertainty did not show statistically significant regression coefficients and were excluded from the model.

Once the relevant variables to predict the marginalization of HP and their relationships were identified, these were integrated into a single model through path analysis. All trajectories show statistically crucial Critical Ratio (CR) values. As shown in Fig. 2, the model explains 53% of the variance of marginalization towards HP, where the risk of infection of HP (CR = 14.02, $p$ < .001) and selfishness (CR = 6.56, $p$ < .001) increase the odds of marginalization, while positive beliefs towards HP decrease them (CR = −2.59, $p$ = .009). In turn, 9% of the variance in the risk of contagion of HP is explained, derived from positive effects of selfishness (CR = 3.79, $p$ < .001), from the uncertainty due to the COVID-19 pandemic (CR = 2.49, $p$ = .01) and negative trust in health institutions (CR = −2.72, $p$ = .007). The variance of positive beliefs towards HP is explained by 7%, derived from the positive effects of trust in institutions (CR = 2.95, $p$ = .003) and perceived selfishness (CR = 2.26, $p$ = .02), and negative effects of the perceived risk of contagion of HP (CR = −2.24, $p$ = .02). Finally, the variance of selfishness is explained by 8%, originating solely from selfishness perceived in others during the pandemic (CR = 5.10, $p$ < .001).

The indicator Chi$^2$ = 15.67, $df$ = 10, $p$ = .10 shows that the discrepancies between the relationships established in the model and those observed in the data matrix are not statistically significant. With a value of SRMR = .04, it can be assumed that the model has a tolerable level of residual variance once the trajectories have explained the variance of the dependent variables. Being above .95 and .90, respectively, the GFI = .98 and AGFI = .95 values indicate that the variance explained by the model is generally adequate. The CFI = .98 indicator tells us that the fit of the model is significantly better than the fit of a null relationship model. The indicator RMSEA =.04, CI 90% [< .001, .08], PCLOSE=.53 indicates that we could expect an equally good fit for this model when replicated in other samples from the same population. Overall, these indicators indicate adequate goodness of fit (*Kline, 2016*).

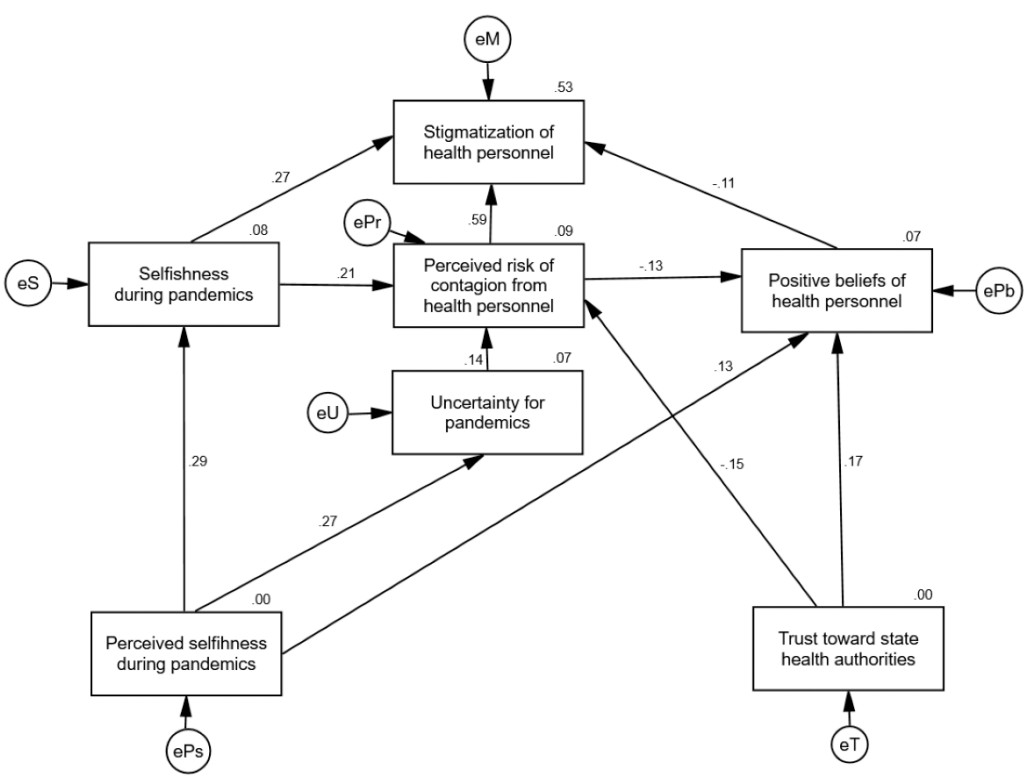

**Figure 2** **Path analysis to explain the disposition to stigmatize health personnel.** The path analysis explains 53% of the variance of stigmatiization towards healthcare personnel, showing indicators of adequate goodness of fit. Standardized values are shown. Source: Own elaboration.

## DISCUSSION

No case of violence is acceptable, but fortunately, so far only isolated cases of violence have been observed in Mexico, and no case, at least known, of lynching or more extreme forms of violence that cost the lives of HP have been identified during the pandemic. This coincides with the results presented here of low disposition to marginalization in the measurements of both studies. However, the fact that there are minorities that report high scores in this measurement should not be neglected. Although they are few, it must be considered that acts of extreme social violence require only some committed inciters to ignite an entire community fearful for its safety and lead it to commit inhumane acts of violence through processes of social contagion (*Bonnasse-Gahot et al., 2018*).

Nevertheless, the interpretation of these levels of stigmatization should be interpreted considering the analyzed samples are not representative of the Mexican population as a whole. Even though the study has large samples from different regions of Mexico, the sampling strategy was limited by the available resources of the project and did not allow a data distribution that would representatively cover the different regions of the country.

In the model, the effects of uncertainty and selfish strategies generated by the pandemic that increase the marginalization of HP should be highlighted. This reaction can be

explained because of the in-group bias (*Hewstone, Rubin & Willis, 2002*), which is a strategy aimed at seeking stable reciprocal links that encourage trust towards and cooperation with those who are perceived as members of the group itself, seeking to reduce the risk of being betrayed by members of other groups who do not share the same interests (*Yamagishi & Kiyonari, 2000*). This bias does not necessarily imply hostility towards members of other groups (*Brewer, 1999*), but *Choi & Bowles (2007)* have proposed that this hostility (known as parochialism) and ingroup bias have evolved together in our species as strategies to appropriate scarce resources essential for survival (*Grossman & Mendoza, 2003*). These results are also congruent with the behavioral immune system model (*Schaller, 2015*), in which the individuals of a community would seek to isolate themselves from members of other groups that imply a potential risk, whether from contagion or competition for scarce resources.

This perception of HP as *others*, outside of the community, could also be explaining the inability of social capital to reduce marginalization. Social capital could reduce the marginalization of members of their community, but not necessarily of people outside of it. In fact, the results of *Alcorta et al. (2020)* show that social capital is a facilitator for achieving community goals, which are not always peace-oriented. In reference to their study conducted in Africa, they note that a strong identity with the community is associated with a greater disposition to political violence, where social capital would serve as a catalyst for actions against other groups perceived as different.

This pandemic has exposed a risk of marginalization that seems new to most HP, although it has been a constant experience for those fighting ancient endemic contagious diseases such as malaria, Ebola, or leprosy. For instance, the meta-analysis of *Yuan et al. (2021)* shows that stigmatization towards HP has been present before the emergence of different pandemics in various regions of the world, especially in middle or lower-income communities or with low levels of education.

These experiences make it necessary to reflect on the integration of healthcare centers and their staff in the communities they serve, as part of that same social fabric, for which community interventions and the collaboration of health units with other local authorities would be necessary. This integration would favor a common identity for the inhabitants and HP, which would reduce the risk of marginalization, but would also facilitate other prevention processes if they would be perceived as people interested in achieving good for the community, namely, *their* community. If achieved, this integration would also favor trust in health authorities, an element that is identified here as relevant for improving the perception of HP.

## CONCLUSIONS

The scores of marginalization and perceived risk of contagion are low, while the scores of positive beliefs are high, indicating a general positive perception of HP. The main predictor of marginalization is the perceived risk of contagion, which is increased by the strategy of selfishness and the uncertainty generated by the COVID-19 pandemic. Social capital does not contribute to preventing the marginalization of HP. Strategies of selfishness, contrary

to cooperation, motivate the marginalization of HP and increase the risk of contagion perceived in HP. Confidence in the state health authorities reduces the perceived risk of contagion and promotes positive beliefs towards HP, making clear the importance of the authorities to prevent marginalization and their ability to support their personnel from the confidence that their work generates in communities. The perceived susceptibility of contagion was not relevant to predicting marginalization or antecedent factors such as personal selfishness or the risk of contagion of HP, indicating that these factors can be explained by the high risk perceived in others, and not in one's own vulnerability.

## ACKNOWLEDGEMENTS

We thank all the people who participated in this study and to the reviewers, whose comments improved the quality of the manuscript.

### Funding
The authors received no funding for this work.

### Competing Interests
The authors declare there are no competing interests.

### Author Contributions
- Christian Enrique Cruz-Torres conceived and designed the experiments, performed the experiments, analyzed the data, prepared figures and/or tables, authored or reviewed drafts of the article, and approved the final draft.
- Jaime Martín del Campo-Ríos conceived and designed the experiments, performed the experiments, authored or reviewed drafts of the article, and approved the final draft.

### Human Ethics
The following information was supplied relating to ethical approvals (i.e., approving body and any reference numbers):

The Autonomous University of Juárez City granted full ethical approval to conduct the study (Ethical Permission Reference: CEI-2020-2-43).

### Data Availability
The raw measurements are available in the Supplementary Files.

### Supplemental Information
Supplemental information for this article can be found online at http://dx.doi.org/10.7717/peerj.14503#supplemental-information.

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
