# Peer review of "Stigmatization towards healthcare personnel during the first COVID-19 wave in Central and Northern Mexico"

_PeerJ, doi:10.7717/peerj.14503_

## Round 0.1 · original submission · Major Revisions

Dear authors,

Please respond point by point to reviewers' comments and make appropriate revisions.

Reviewer 1 ·

Basic reporting

1. Abstract: you should start with a first paragraph describing the background.

2. The theoretical framework is scarce, you should clearly describe the scientific evidence that supports the hypothesis you have raised.

3. A lot of necessary information is missing in methods section:
- Experimental procedures should be better defined
- More information should be provided about the participants’ characteristics.
- You should better defined inclusion and exclusion criteria

4. The Discussion should be enriched with the existing theory. You should clearly describe the scientific evidence that supports your findings.

5. The references are correct but weak and incomplete, thus they should be enriched. Moreover, it would be appropriate to include the DOI to all references.

Experimental design

no comment

Validity of the findings

no comment

Additional comments

no comment

Reviewer 2 ·

Basic reporting

The introduction needs more detail. About marginalization there is only one reference. It is recognized the value from quoting study cases, but it is necessary to realize an exhaustive literature review about marginalization to build a solid theory framework. For example: Which elements difference between marginalization and discrimination or segregation or labor precarization?

Experimental design

The study’s objective is exploring the disposition to marginalize healthcare personnel (HP) in Mexico during the COVID-19 pandemic. But the survey from which the data was collected it was conducted during the second and fourth week of April 2020. This period corresponds only to first wave of COVID-19 pandemic. Nowadays (September 2022) Mexico is living the fifth wave the COVID-19 pandemic.

In introduction indicates 520 participants responded to three instruments to measure the disposition to marginalization. Also, it is indicating 76% participants were residents of northern and 24% from central-southern states of Mexico. Whit these elements the authors concluding that in Mexico it is a Marginalization towards healthcare personnel.

In a country with 126 million of people and a major concentration of people in states of center, it doesn´t allow to accept a sample so few and territorial biased to northern states. It is necessary to use a sample method as well as determinate the error value and confidence level expected.

Validity of the findings

Thus, results of the Chrombach's Alpha test are valid, but the sample size and the lack of a territorial sampling design do not allow measuring the effect on a national scale. Accepting the results would be accepting a fallacy of exception. The results only serve to validate that particular test.

Additional comments

On another hand, the document contains two studies. Study 1. Descriptive and sociodemographic components of marginalization; and Study 2: Psychosocial predictors of marginalization towards healthcare personnel. After reading them there is a feeling that both studies were made by separated and then they got together. It is necessary to conform to the structure for submitting articles: introduction, literature review, methodology (data, statistical procedure), results presentation, conclusion.

As to references in text it protrudes one document that is referring eight times. The bibliography contains 32 documents and juts one of them have eight references. Also, there are a couple documents aren´t reference in text.

---

## Round 0.2 · accepted · Accept

Dear authors,

Congratulations.

The manuscript has greatly improved after carefully following the reviewers' suggestions.